# Health and cost impact of stepping down asthma medication for UK patients, 2001–2017: A population-based observational study

**Chloe I. Bloom**[1]*, **Laure de Preux**[2], **Aziz Sheikh**[3], **Jennifer K. Quint**[1]

**1** National Heart and Lung Institute, Imperial College London, London, United Kingdom, **2** Imperial College Business School, Imperial College London, London, United Kingdom, **3** Asthma UK Centre for Applied Research, Usher Institute, University of Edinburgh, Edinburgh, Scotland, United Kingdom

* chloe.bloom06@imperial.ac.uk

**Data Availability Statement:** Data are available on request from the Clinical Practice Research Datalink (CPRD). Their provision requires the

## Abstract

### Background

Guidelines recommend stepping down asthma treatment to the minimum effective dose to achieve symptom control, prevent adverse side effects, and reduce costs. Limited data exist on asthma prescription patterns in a real-world setting. We aimed to evaluate the appropriateness of doses prescribed to a UK general asthma population and assess whether stepping down medication increased exacerbations or reliever use, as well as its impact on costs.

### Methods and findings

We used nationwide UK primary care medical records, 2001–2017, to identify 508,459 adult asthma patients managed with preventer medication. Prescriptions of higher-level medication: medium/high-dose inhaled corticosteroids (ICSs) or ICSs + add-on medication (long-acting β2-agonist [LABA], leukotriene receptor antagonist [LTRA], theophylline, or long-acting muscarinic antagonist [LAMA]) steadily increased over time (2001 = 49.8%, 2017 = 68.3%). Of those prescribed their first preventer, one-third were prescribed a higher-level medication, of whom half had no reliever prescription or exacerbation in the year prior. Of patients first prescribed ICSs + 1 add-on, 70.4% remained on the same medication during a mean follow-up of 6.6 years. Of those prescribed medium/high-dose ICSs as their first preventer, 13.0% already had documented diabetes, cataracts, glaucoma, or osteopenia/osteoporosis. A cohort of 125,341 patients were drawn to assess the impact of stepping down medication: mean age 50.4 years, 39.4% males, 39,881 stepped down. Exposed patients were stepped down by dropping their LABAs or another add-on or by halving their ICS dose (halving their mean-daily dose or their inhaler dose). The primary and secondary outcomes were, respectively, exacerbations and an increase in reliever prescriptions. Multivariable regression was used to assess outcomes and determine the prognostic factors for initiating stepdown. There was no increased exacerbation risk for each possible medication stepdown (adjusted hazard ratio, 95% CI, p-value: ICS inhaler dose = 0.86, 0.77–0.93, p < 0.001; ICS mean daily = 0.80, 0.74–0.87, p < 0.001; LABA = 1.01, 0.92–1.11, p = 0.87, other

purchase of a license and our license does not permit us to make them publicly available to all. We used data from the version collected in January 2018 and have clearly specified the data selected in our Methods section to allow identical data to be obtained by others, via the purchase of a license, using the published code lists and those included in the supplementary. Licences are available from the CPRD (http://www.cprd.com): The Clinical Practice Research Datalink Group, The Medicines and Healthcare products Regulatory Agency, 10 South Colonnade, Canary Wharf, London, UK.

**Funding:** The authors received no specific funding for this work.

**Competing interests:** I have read the journal's policy and the authors of this manuscript have the following competing interests. CIB reports financial support outside the submitted work from Asthma UK, Chiesi, and AstraZeneca. AS reports that his institution received research support from Asthma UK for the Asthma UK Centre for Applied Research, outside the submitted work. AS is a member of the Editorial Board of *PLOS Medicine*. JKQ reports grants from MRC, GSK, BLF, Asthma UK, The Health Foundation, IQVIA, AstraZeneca, Chiesi, BI, Bayer; personal fees from AZ, Chiesi, BI, Bayer, and TEVA, outside the submitted work. LDP declares no competing interests associated with this manuscript.

**Abbreviations:** BMI, body mass index; BTS, British Thoracic Society; COPD, chronic obstructive pulmonary disease; CPRD, Clinical Practice Research Datalink; GINA, Global Initiative for Asthma; HES, Hospital Episode Statistics; ICS, inhaled corticosteroid; ISAC, Independent Scientific Advisory Committee; LABA, long-acting beta-agonist; LAMA, long-acting muscarinic antagonist; LTRA, leukotriene receptor antagonist; NHS, National Health Service; ONS, Office for National Statistics; RCT, randomised control trial; SIGN, Scottish Intercollegiate Guidelines Network.

add-on = 1.00, 0.91–1.09, p = 0.79) and no increase in reliever prescriptions (adjusted odds ratio, 95% CI, p-value: ICS inhaler dose = 0.99, 0.98–1.00, p = 0.59; ICS mean daily = 0.78, 0.76–0.79, p < 0.001; LABA = 0.83, 0.82–0.85, p < 0.001; other add-on = 0.86, 0.85–0.87, p < 0.001). Prognostic factors to initiate stepdown included medication burden, but not medication side effects. National Health Service (NHS) indicative prices were used for cost estimates. Stepping down medication, either LABAs or ICSs, could save annually around £17,000,000 or £8,600,000, respectively. Study limitations include the possibility that prescribed medication may not have been dispensed or adhered to and the reason for stepdown was not documented.

## Conclusion

In this UK study, we observed that asthma patients were increasingly prescribed higher levels of treatment, often without clear clinical indication for such high doses. Stepping down medication did not adversely affect outcomes and was associated with substantial cost savings.

## Author summary

### Why was this study done?

- There is a growing interest in rationalising medicines through deprescribing.

- Specifically, asthma guidelines strongly recommend stepping down asthma medication in stable patients to reduce the risk of side effects from long-term use, in particular from inhaled corticosteroid use.

- Guideline recommendations are based on trials, comparing stable patients that were stepped down to those that were not, but these trials were small in size, had short follow-up periods, and may not represent what happens in day-to-day clinical practice.

### What did the researchers do and find?

- We studied adult asthma patients from across the UK, using primary care electronic medical records, and found prescriptions of higher-dose asthma medication have steadily increased between 2001 and 2017.

- Stepping down medication occurred infrequently but did not increase asthma exacerbations or the use of reliever medication; patients at risk of medication side effects were not preferentially stepped down.

- This clinical practice was shown to have considerable cost savings if carried out in stable asthma patients.

### What do these findings mean?

- There is an increasing tendency to prescribe higher-dose, expensive asthma medications.

- Stepping down was found to be safe and highly cost-effective and may reduce long-term medication adverse effects.

## Introduction

Over the past 4 decades, the use of inhaled corticosteroids (ICSs) and, subsequently, long-acting β2-agonists (LABAs) have resulted in extraordinary improvements in outcomes for people with asthma [1,2]. Since the mid-1990s, asthma guidelines and strategies (including Global Initiative for Asthma [GINA], National Asthma Education and Prevention Program Expert Panel Report, and British Thoracic Society/Scottish Intercollegiate Guidelines Network [BTS/SIGN]) have offered a coherent, evidenced-based, stepwise approach for pharmacological management that can be used within primary and specialist care settings [3–5]. These guidelines strongly advocate finding the minimum effective treatment dose that can achieve symptom control. Indifference to this recommendation increases the risk of serious adverse medication effects because of the chronicity of treatment and unquestionably increases medication costs. The ongoing improvement in asthma care, alongside a higher disease prevalence and rapid growth of more expensive newer drugs, has led to ever-increasing total medication costs across many countries [6–8]. For example, in the UK, asthma costs are over £1.1 billion, of which around 80% is for drug costs, and in the United States, the cost of asthma is around $80 billion annually, of which nearly two-thirds is for prescriptions [6].

The GINA 2019 guidelines recommend a paradigm shift in asthma treatment, advising all asthma patients to receive ICSs, starting with low-dose ICSs or as-needed low-dose ICS–formoterol in the GINA guidelines, and escalating as necessary. For the majority of patients, 80%–90% of the therapeutic benefit of ICSs are obtained with low doses [9,10]. Prolonged use and higher doses of ICS are associated with a progressive risk of systemic adverse effects, including adrenal suppression, diabetes, cataracts, glaucoma, osteoporosis, and fractures. Unlike the plateau effect of efficacy outcomes, which occurs with higher ICS doses, there is no plateau to the risk of adverse effects [11]. Community studies from Australia and Scotland suggest patients are often inappropriately prescribed, and remain on, high doses of ICSs [12,13]. International and national guidelines recommend stepping down treatment once asthma is controlled to prevent accumulating adverse effects, yet it is thought to be an uncommon practice [3,14], even though the majority of asthma patients seldom exacerbate [15].

In this study, using nationally representative primary care data from across the UK, we first evaluated the temporal pattern of preventer prescriptions (prevalent and incident prescriptions), the associated patient characteristics, and the frequency of medication step change. We then focused on patients that were stepped down to evaluate any associated health and cost impact.

## Methods

### Ethical approval

The protocol for this research was approved by the Independent Scientific Advisory Committee (ISAC) for MHRA Database Research (protocol number 18_120) and is available alongside the STROBE checklist for observational studies (S1 STROBE Checklist & S1 Study Protocol). This study is based, in part, on data from the Clinical Practice Research Datalink (CPRD) obtained under licence from the UK Medicines and Healthcare products Regulatory Agency.

The data are provided by patients and collected by the National Health Service (NHS) as part of their care and support. Linked pseudonymised data were provided for this study by CPRD. Data are linked by NHS Digital, the statutory trusted third party for linking data, using identifiable data held only by NHS Digital. Select general practices consent to this process at a practice level, with individual patients having the right to opt-out. The Office for National Statistics (ONS) was the provider of the ONS data contained within the CPRD data. The interpretation and conclusions contained in this study are those of the authors alone.

## Data sources

We used the CPRD-GOLD, a nationally representative database of deidentified UK primary care electronic medical records. CPRD holds information on diagnoses, symptoms, and prescriptions on more than 11 million patients [16]. It is one of the largest longitudinal healthcare databases worldwide and has been validated extensively [16]. Secondary care information was obtained from the Hospital Episode Statistics (HES) database. HES only covers English NHS hospitals, so around 60% of CPRD practices have individual level HES linkage, socioeconomic data (using the Index of Multiple Deprivation), and mortality data (ONS).

In the UK, each single inhaler prescribed should last one month if taken as per manufacturer intended dose (usually as 2 puffs twice a day). A new prescription is recorded for every successive inhaler, and more than one of the same inhaler can be prescribed on the same day (and is recorded as such).

## Study populations and design

**Main study population.** Inclusion criteria were an asthma diagnosis using a validated algorithm of including asthma clinical codes (with an 86% positive predictive value) [17], no chronic obstructive pulmonary disease (COPD) co-diagnosis [18], ≥18 years old, and ≥1 year of prescription data (Fig 1 & Fig 2). Prescription data were considered eligible for study inclusion ('eligible prescription date') from the latest date of the following: January 1st, 2001, research acceptable date (CPRD quality control), continuous records date, 18th birthday, or asthma diagnosis date. Follow-up was censored at the earliest date of the following: January 1st, 2018, date transferred out of CPRD, last data collection, or death.

**Prescription analysis (prevalent and incident prescriptions).** To be eligible, patients had to have ≥1 year of follow-up and ≥3 preventer prescriptions (Figs 1 & S1). For the incident (first asthma prescription) analysis, patients were included if they had ≥2 asthma prescriptions within the year after their incident prescription (to exclude patients prescribed a 'trial' inhaler; in the UK, trialling an inhaler is often part of the diagnostic pathway to determine whether a patient has asthma or not), ≥1 year of no preventer prescriptions before their incident prescription, and were HES-linked. Patients characteristics in the incident analysis were described to assess their suitability for their level of inhaler and their potential risk of medication adverse effects. A change in prescriptions was described using the 2016 SIGN/BTS stepwise approach [3].

**Stepping-down outcomes analysis.** To determine the health impact, a cohort of regular preventer users was drawn from the main study population (Fig 1 & S1 Fig). Exposed patients had asthma medication stepped down, and unexposed patients were all other eligible patients. Patients could only enter this cohort once they had been prescribed ≥3 ICSs during a 1-year period after their eligible prescription date and were HES-linked. This cutoff was chosen to include patients with regular inhaler use, potentially suitable for reduction in their treatment level. The 'step-down date' was defined as either the date of stepping down (exposed patients)

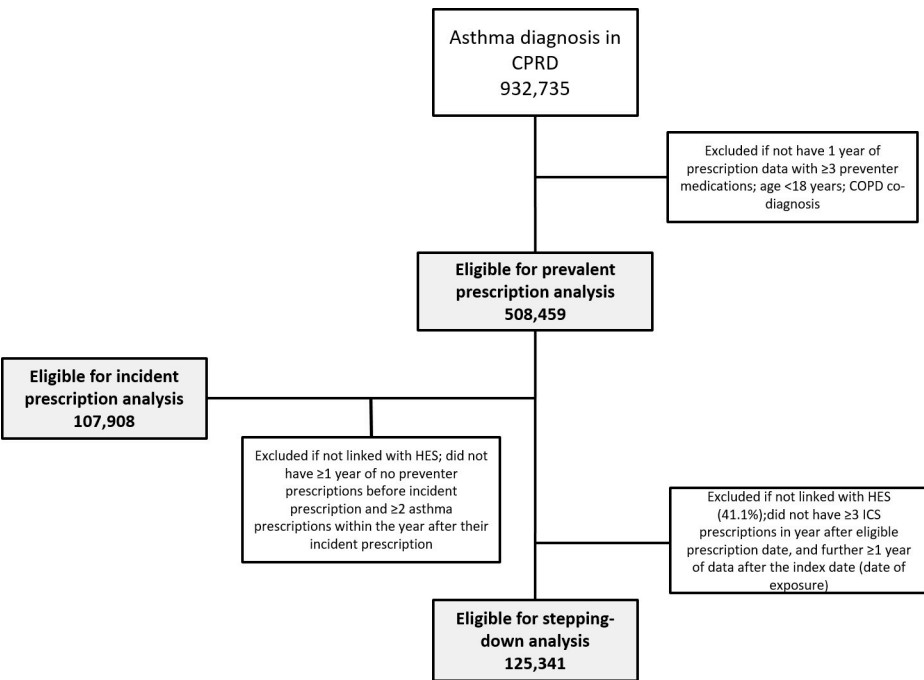

**Fig 1. Study design.** COPD, chronic obstructive pulmonary disease; CPRD, Clinical Practice Research Datalink; HES, Hospital Episode Statistics; ICS, inhaled corticosteroid

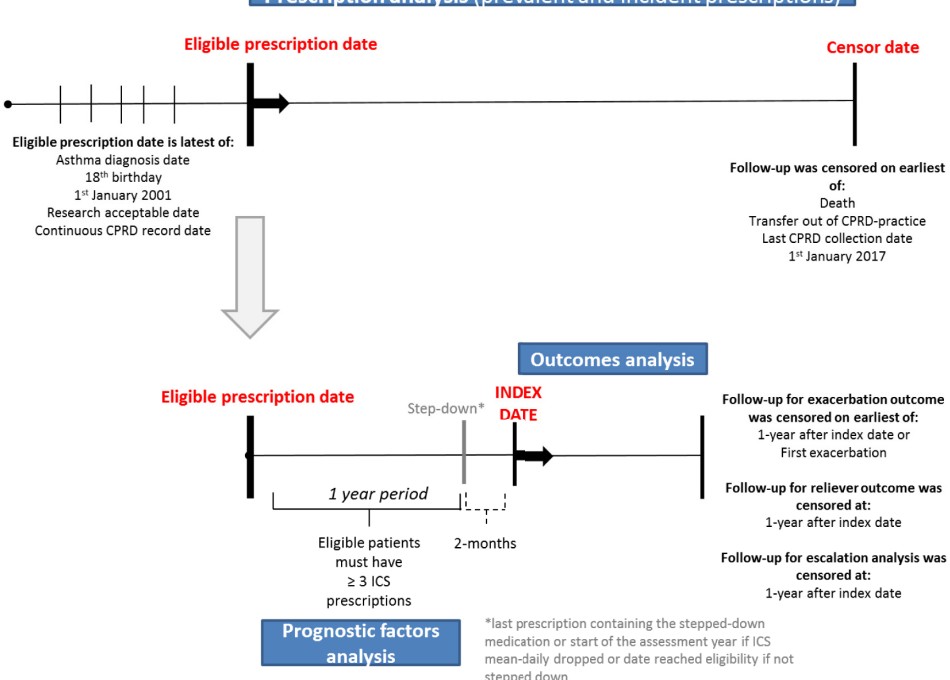

**Fig 2. Flow diagram for study.** CPRD, Clinical Practice Research Datalink; ICS, inhaled corticosteroid

or, if unexposed, the first date after satisfying all eligibility criteria. This cohort was also used to determine the prognostic factors associated with instigating stepping down.

The 'index date' for the outcome analysis was then recorded as 2 months from the stepdown date. This time period was set because although a single inhaler prescription should last 1 month, many patients miss doses. In this cohort, patients received a mean of 7 prescriptions per year (IQR 6–9), suggesting each inhaler was used on average for approximately 2 months. Therefore, after this time period, patients were considered to be no longer using their dropped medication; however, a sensitivity analysis was also carried out assuming 1- and 3-month time periods.

## Stepping down and medication escalation

Stepping down was based on guideline recommendations and was defined as (1) dropping an add-on therapy (LABA, leukotriene receptor antagonist [LTRA], theophylline, or long-acting muscarinic antagonist [LAMA]), (2) halving the ICS inhaler dose (for example, beclometasone 200 mcg to 100 mcg), or (3) halving the mean-daily ICS dose (stepping down by reducing daily inhalations) [3,4]. Most ICS stepping-down trials used a 50% stepdown [19]; using a lower stepdown would be less likely to identify an effect if there is one. To be included, patients had to have ≥3 inhalers of the stepped-down drug in the year before stepping down. Only the first stepdown that occurred during follow-up was included.

A drop in mean-daily ICS was measured by comparing the mean ICS dose per day in 1 year with the mean ICS dose per day in the first 4 months of the subsequent year ('assessment year'). The first day of the assessment year was recorded as the index date.

Escalation was defined as the opposite of stepping down (i.e., addition of an add-on therapy or doubling of ICS dose) occurring in the year following the index date.

## Outcomes and confounders

The primary outcome was asthma exacerbations. These were identified as previously defined [20], as a short course of oral corticosteroids, an emergency department visit for asthma, a hospital admission, or death secondary to asthma. An exacerbation had to occur within 12 months of the patient's study entry date. The secondary outcome was a change in reliever prescriptions. This was defined as an increase of ≥1 reliever prescription (short-acting beta-agonist) in the year after the index date compared with the year before. Frequency of SABA is used to help assess asthma control and is often used in observational studies as a proxy for asthma control [21–24]. A history of atopy, gastroesophageal reflux, smoking, anxiety, depression, cataracts, pneumonia, diabetes, osteopenia, osteoporosis, and arrhythmia was recorded using appropriate Read codes (see S1 CPRD_medcodes). Variables defined only using data from the year prior to the index date were 'ICS stability' (binary variable; defined as change or no change in ICS inhaler prescription dose or type), 'maximum ICS dose' (categorical variable [low, medium, or high]); maximum dose of ICS prescription), 'ICS frequency' (categorical variable (0–3, 4–6, 7–10, ≥11); number of ICS prescriptions) and reliever frequency (binary variable (<3, ≥3); number of short-acting beta-agonist canisters). Also, using only data from the previous year, nonpharmacological management was assessed using appropriate Read codes for the occurrence of the following within primary care: patient given an asthma management plan, annual asthma review, or inhaler technique check. Patients prescribed LABA–ICS were categorised by combination inhaler or not (this variable was included as a secondary analysis, after a reviewer's suggestion). ICS dose was categorised using the 2016 SIGN/BTS guidelines, based on fine-dose beclometasone dipropionate equivalent: low dose (≤799 mcg), medium dose (800–1,599 mcg), and high dose (≥1,600 mcg) [3]. The cutoff for an infrequent ICS

prescription was defined as ≤2 prescriptions per year, based on the distribution across the whole cohort.

## Statistical analysis

For prevalent prescriptions, the maximum therapy level per patient each calendar year was determined. To evaluate change in prescriptions from the first (incident) preventer medication prescribed, a single prescription increase/decrease during a patient's total follow-up was recorded as a change. The effect of stepping down on exacerbations was analysed in an intention-to-treat Cox proportional hazards model adjusted for sex, age, body mass index (BMI), smoking, socioeconomic class, gastroesophageal reflux, anxiety, depression, annual asthma review, inhaler technique check, asthma management plan, exacerbations, reliever use, maximum ICS dose, ICS frequency, ICS stability, and add-on therapies using complete case analysis. Patients were censored on their first exacerbation or 1 year after the index date. Schoenfeld residuals were found not to violate the proportional hazards assumption [25]. The effect of stepping down on reliever use was analysed in an intention-to-treat logistic regression model adjusted for the same variables as the Cox model. The association between potential prognostic factors and initiating stepdown was assessed using multivariable mixed-effects logistic regression models, using GP practice as the random intercept. Several sensitivity and subgroup analyses were performed (1) using a 6-month (instead of 12-month) outcome window after the index date, (2) defining the date of stepping down at 1 month and at 3 months after the last prescription date (instead of 2 months), (3) stratifying patients by presence of exacerbations in the year prior to the index date, and (4) analysing the effect on exacerbations using an as-treated approach (censoring stepped-down patients on date of escalation, if within 1 year of the index date). To further diminish the effect of selection bias, an additional analysis using inverse-probability–weighted propensity score methodology was used; the average treatment time for the first exacerbation was compared between each stepdown and no stepdown. All statistical analyses were performed using STATA 14.2.

## Cost analysis

Cost analyses for the cohort were performed applying drug costs alone. Price per drug was obtained using 2019 indicative prices from the NHS Dictionary of Medicines and Devices to avoid a cost effect due to price changes [26]. Patients prescribed drugs that no longer had a price available were excluded. Individual cost differences were calculated by subtracting the cost of preventer medications prescribed 12 months post-index date from the yearly drug costs pre-index date. The mean of those differences was calculated for the cohort by medication stepped down. The mean cohort costs were used to estimate population-level costs for stepping down LABAs or ICSs using the following assumptions: UK population was 65 million, of whom 80% were adults (based on 2019 ONS data); 7% had asthma [27]; 50% were prescribed LABAs; 40% were prescribed medium- or high-dose ICSs (Fig 1); 15% had a 'stable' year; and only half of the eligible population were stepped down (Table 1). A 'stable' year was defined as ≥3 ICS prescriptions, no change in preventer medications, or <3 reliever prescriptions and no exacerbations.

## Results

### Prevalent preventer prescriptions

508,459 patients were identified; the proportion on higher-level preventer medications (medium/high-dose ICSs or ICSs and add-on therapy) steadily increased over time (2001 = 49.8%, 2017 = 68.3%; Fig 3).

**Table 1. Change of preventer medication during follow-up from incident prescription.**

|  | Mean Follow-up (95% CI) | No Change (%) | Escalate Only (%) | Stepdown Only (%) | Escalate and Stepdown (%) |
|---|---|---|---|---|---|
| **Low-dose ICSs** | 6.6 years (6.5–6.6) | 41.7 | 39.5 | * | 18.9 |
| **Medium/high-dose ICSs** | 8.1 years (8.0–8.2) | 19.5 | 34.3 | 14.5 | 31.7 |
| **ICSs + 1 add-on** | 6.6 years (6.6–6.7) | 70.4 | 6.3 | 7.8 | 15.5 |

*Patients that stopped all preventer medications were not included. **Abbreviations:** ICS, inhaled corticosteroid.

In 2017, only 31.7% of patients were treated with low-dose ICSs alone. Most patients were managed with ICSs and 1 add-on. Of those prescribed an add-on, 88.3% were prescribed a combination LABA–ICS inhaler, of which 69.6% included medium- or high-dose ICSs.

## First-time (incident) preventer prescriptions

107,908 patients were identified for the incident analysis; around one-third were prescribed higher-level medication as their first asthma preventer medication (S1 Fig). Most patients prescribed an add-on were prescribed LABAs (94.0%), of whom 74.6% were prescribed either medium- or high-dose ICSs. Of those prescribed LABA–ICS, 77.2% were prescribed an LABA–ICS combination inhaler throughout the time period, but by 2016 onwards, all LABA–ICS prescriptions were combination inhalers.

In patients prescribed ICSs + 1 add-on, 70.4% did not have their prescription changed during a mean follow-up of 6.6 years (Table 1). Of those prescribed medium- or high-dose ICSs, 54.5% kept the same therapy or escalated during a mean follow-up of 8.1 years.

Of those prescribed higher-level medication as their incident asthma prescription (all steps above low-dose ICSs), 48.4% had no exacerbation or reliever prescriptions in the previous year. The median time since diagnosis before first prescription was 19.2 months, and 13.2% already had a comorbidity that could be worsened by corticosteroids (S1 Table).

Of those prescribed medium- or high-dose ICSs as their incident prescription, 13.0% had known diabetes, cataracts, glaucoma, osteopenia, or osteoporosis at the time of prescription, compared with 11.2% prescribed incident low-dose ICSs (Table 2).

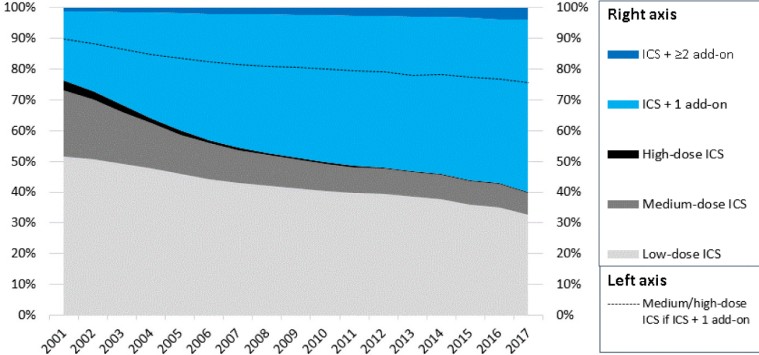

**Fig 3. Prevalent asthma preventer prescriptions from across the UK, 2001–2017.** The light blue area below the dotted line is the percentage of patients prescribed medium-dose or high-dose ICSs that are prescribed an ICS and 1 add-on therapy. Add-on therapy refers to LABA, LTRA, theophylline, or LAMA. ICS, inhaled corticosteroid; LABA, long-acting β2-agonist; LAMA, long-acting muscarinic antagonist; LTRA, leukotriene receptor antagonist.

**Table 2. Patient characteristics at their first asthma preventer prescription by ICS dose prescribed; low-dose ICSs compared with medium/high-dose ICSs.**

| | Low-Dose ICSs Incident Prescription | Medium/High-Dose ICSs Incident Prescription |
|---|---|---|
| | Median (IQR) | Median (IQR) |
| Age (years) | 48 (35–62) | 52 (39–64) |
| Reliever frequency | 1 (0–2) | 0 (0–1) |
| | N (%) | N (%) |
| Total | 68,363 (100%) | 39,545 (100%) |
| Never smoked | 22,727 (33.2%) | 12,186 (30.8%) |
| Atopy | 32,314 (47.3%) | 18,544 (46.9%) |
| Anxiety | 13,016 (19.0%) | 7,455 (18.9%) |
| Reflux | 6,993 (10.2%) | 4,746 (12.0%) |
| Bronchiectasis | 306 (0.4%) | 418 (1.1%) |
| GP-treated exacerbations | 7,666 (11.2%) | 5,379 (13.6%) |
| Hospital exacerbations | 883 (1.3%) | 885 (2.2%) |
| IHD | 3,181 (4.7%) | 2,502 (6.3%) |
| Hypertension | 14,211 (20.8%) | 9,073 (22.9%) |
| Arrhythmia | 2,021 (3.0%) | 1,536 (3.9%) |
| Pneumonia | 1,812 (2.7%) | 1,332 (3.4%) |
| Comorbidity that could be worsened by corticosteroids | 7,674 (11.2%) | 5,128 (13.0%) |
| Glaucoma | 862 (1.3%) | 639 (1.6%) |
| Cataracts | 2,125 (3.1%) | 1,419 (3.6%) |
| Osteopenia/osteoporosis | 1,960 (2.9%) | 1,337 (3.4%) |
| Diabetes | 3,875 (5.7%) | 2,498 (6.3%) |

**Abbreviations:** GP, general practitioner; ICS, inhaled corticosteroid; IHD, ischaemic heart disease.

## Effect of stepping down on exacerbations

125,341 patients (mean age 50.4 years, 39.4% males) satisfied the criteria for the stepping-down analysis: 39,881 were stepped down (ICS mean daily = 26,603, ICS inhaler dose = 6,182, LABA = 4,078, other add-on = 3,018), and 85,460 were not stepped down (Table 3).

Stepping down LABA or other add-on medication or ICSs, either by reducing the dose of the inhaler or by reducing the number of puffs, did not significantly increase the risk of an exacerbation in the subsequent year (adjusted hazard ratio, 95% CI, p-value: ICS inhaler dose = 0.86, 0.77–0.93, $p < 0.001$; ICS mean daily = 0.80, 0.74–0.87, $p < 0.001$; LABA = 0.99, 0.92–1.11, $p = 0.87$; other add-on = 0.99, 0.91–1.09, $p = 0.79$; Table 4). The main findings remained robust in the sensitivity analyses (S2 Table). The stratified analysis by prior exacerbation history showed no significant difference (S3 Table). The treatment effect analysis found negligible difference in days between first exacerbation in patients with stepped-down drug and no stepdown (S4 Table).

The proportion of unexposed patients that had treatment escalated in the year following the index date was similar to or higher than that for mean-daily ICS or LABA (unexposed = 16.7% [95% CI 16.3–17.0], mean-daily ICS = 16.3% [95% CI 15.8–16.7]; LABA = 11.0% [95% CI 10.1–12.0]; S5 Table). Patients that dropped their ICS inhaler dose, LTRA, LAMA, or theophylline had higher proportions escalate (ICS inhaler dose = 29.5% [95% CI 28.3–30.7], LTRA = 20.5% [95% CI 18.7–22.4], LAMA = 23.2% [95% CI 20.0–26.5], theophylline = 59.8% [95% CI 55.9–63.7]).

**Table 3. Patient characteristics of each exposed and unexposed group.**

| Characteristics | Controls/Unexposed | | Stepped-Down Drug | | | | | |
|---|---|---|---|---|---|---|---|---|
| | Not Stepped Down | | ICSs | | LABA | | Other Add-on | |
| | N | % | N | % | N | % | N | % |
| Total | 85,460 | 100.0 | 32,785 | 100.0 | 4,078 | 100.0 | 3,018 | 100.0 |
| **ICS stepdown** | | | | | | | | |
| Mean-daily ICS | – | – | 26,603 | 81.1 | – | – | – | – |
| Inhaler dose | – | – | 6,182 | 18.9 | – | – | – | – |
| **Other add-on** | | | | | | | | |
| LTRA | – | – | . | . | – | – | 1,751 | 58.0 |
| Theophylline | – | – | . | . | – | – | 605 | 20.0 |
| LAMA | – | – | . | . | – | – | 662 | 21.9 |
| **Age** | | | | | | | | |
| 18–29 | 10,936 | 12.8 | 4,503 | 13.7 | 522 | 12.8 | 273 | 9.0 |
| 30–40 | 13,303 | 15.6 | 5,899 | 18.0 | 610 | 15.0 | 418 | 13.9 |
| 40–50 | 16,403 | 19.2 | 6,984 | 21.3 | 710 | 17.4 | 594 | 19.7 |
| 50–60 | 14,603 | 17.1 | 5,691 | 17.4 | 661 | 16.2 | 624 | 20.7 |
| 60–70 | 14,367 | 16.8 | 4,817 | 14.7 | 673 | 16.5 | 537 | 17.8 |
| ≥70 | 15,848 | 18.5 | 4,891 | 14.9 | 902 | 22.1 | 572 | 19.0 |
| **Males** | 34,100 | 39.9 | 13,049 | 39.8 | 1,465 | 35.9 | 1,020 | 33.8 |
| **Socioeconomic status** | | | | | | | | |
| 1 (least deprived) | 13,920 | 16.3 | 4,884 | 14.9 | 637 | 15.6 | 411 | 13.6 |
| 2 | 16,347 | 19.1 | 6,555 | 20.0 | 811 | 19.9 | 593 | 19.6 |
| 3 | 16,154 | 18.9 | 6,565 | 20.0 | 779 | 19.1 | 539 | 17.9 |
| 4 | 18,041 | 21.1 | 7,191 | 21.9 | 881 | 21.6 | 632 | 20.9 |
| 5 | 20,998 | 24.6 | 7,590 | 23.2 | 970 | 23.8 | 843 | 27.9 |
| **BMI** | | | | | | | | |
| Normal | 18,120 | 23.0 | 7,350 | 24.0 | 873 | 22.7 | 630 | 21.9 |
| Underweight | 1,659 | 2.1 | 608 | 2.0 | 86 | 2.2 | 59 | 2.1 |
| Overweight | 32,929 | 41.8 | 12,802 | 41.8 | 1,544 | 40.2 | 1,157 | 40.3 |
| Obese | 26,163 | 33.2 | 9,876 | 32.2 | 1,335 | 34.8 | 1,026 | 35.7 |
| **Smoking history** | | | | | | | | |
| Never | 27,822 | 32.6 | 10,799 | 32.9 | 1,264 | 31.0 | 967 | 32.0 |
| Current | 28,452 | 33.3 | 11,383 | 34.7 | 1,359 | 33.3 | 970 | 32.1 |
| Ex-smoker | 29,186 | 34.2 | 10,603 | 32.3 | 1,455 | 35.7 | 1,081 | 35.8 |
| **Atopy** | 46,104 | 53.9 | 17,917 | 54.6 | 2,283 | 56.0 | 1,842 | 61.0 |
| **Anxiety** | 18,836 | 22.0 | 6,816 | 20.8 | 1,031 | 25.3 | 777 | 25.7 |
| **Depression** | 22,921 | 26.8 | 8,582 | 26.2 | 1,227 | 30.1 | 946 | 31.3 |
| **Reflux** | 11,787 | 13.8 | 4,151 | 12.7 | 717 | 17.6 | 571 | 18.9 |
| **In year prior to exposure** | | | | | | | | |
| ≥1 exacerbation | 8,204 | 9.6 | 2,750 | 8.4 | 526 | 12.9 | 585 | 19.4 |
| **ICS frequency** | | | | | | | | |
| 3 | 18,328 | 21.4 | 6,582 | 20.1 | 937 | 23.0 | 350 | 11.6 |
| 4–6 | 37,611 | 44.0 | 17,467 | 53.3 | 1,924 | 47.2 | 1,123 | 37.2 |
| 7–10 | 19,439 | 22.7 | 6,410 | 19.6 | 833 | 20.4 | 930 | 30.8 |
| ≥11 | 10,082 | 11.8 | 2,326 | 7.1 | 384 | 9.4 | 615 | 20.4 |
| **Max ICS** | | | | | | | | |
| Low dose | 48,136 | 56.3 | 16,266 | 49.6 | 1,647 | 40.4 | 937 | 31.0 |
| Medium dose | 26,969 | 31.6 | 11,295 | 34.5 | 1,669 | 40.9 | 1,010 | 33.5 |

(*Continued*)

**Table 3.** (Continued)

| Characteristics | Controls/Unexposed | | Stepped-Down Drug | | | | | |
|---|---|---|---|---|---|---|---|---|
| | Not Stepped Down | | ICSs | | LABA | | Other Add-on | |
| | N | % | N | % | N | % | N | % |
| High dose | 10,355 | 12.1 | 5,224 | 15.9 | 762 | 18.7 | 1,071 | 35.5 |
| **Stable ICS** | 68,671 | 80.4 | 26,300 | 80.2 | 2,875 | 70.5 | 2,242 | 74.3 |
| **LABA use** | 46,256 | 54.1 | 15,464 | 47.2 | 0 | 0.0 | 2,478 | 82.1 |
| **≥2 add-on therapies** | 686 | 0.8 | 185 | 0.6 | 23 | 0.6 | 528 | 17.5 |
| **Reliever frequency** | | | | | | | | |
| 0–2 | 27,764 | 32.5 | 12,094 | 36.9 | 1,560 | 38.3 | 809 | 26.8 |
| 3–7 | 36,874 | 43.1 | 15,077 | 46.0 | 1,770 | 43.4 | 1,177 | 39.0 |
| ≥8 | 20,822 | 24.4 | 5,614 | 17.1 | 748 | 18.3 | 1,032 | 34.2 |
| **Annual asthma review** | 45,893 | 53.7 | 15,775 | 48.1 | 2,136 | 52.4 | 1,558 | 51.6 |
| **Inhaler technique** | 37,633 | 44.0 | 13,517 | 41.2 | 1,807 | 44.3 | 1,246 | 41.3 |
| **Asthma management plan** | 9,673 | 11.3 | 3,080 | 9.4 | 479 | 11.7 | 297 | 9.8 |

Dashes indicate no data/not applicable. **Abbreviations:** BMI, body mass index; ICS, inhaled corticosteroid; LABA, long-acting β2-agonist; LAMA, long-acting muscarinic antagonist.

## Effect on reliever use

Stepping down was not associated with an increase in reliever use (adjusted odds ratio [95% CI, p-value]: ICS inhaler dose = 0.99 [0.98–1.00, p = 0.59], ICS mean daily = 0.78 [0.76–0.79, p < 0.001], LABA = 0.83 [0.82–0.85, p < 0.001], other add-on = 0.86 [0.85–0.87, p < 0.001]; Table 5).

## Prognostic factors for initiating stepping down

The factors significantly associated with initiating ICS stepdown included high- or medium-dose ICSs, ≥4 ICS prescriptions, current smoker, LABA use, and ICS stability (S6 Table).

**Table 4. Risk of an exacerbation by medication stepped down.**

| | Adjusted HR | 95% CI | p-Value |
|---|---|---|---|
| **Drug stepped down** | | | |
| None | *Reference* | | |
| ICS: inhaler dose | 0.86 | 0.77–0.93 | <0.001 |
| ICS: mean daily | 0.80 | 0.74–0.87 | <0.001 |
| LABA | 0.99 | 0.92–1.11 | 0.871 |
| Other add-on | 0.99 | 0.91–1.09 | 0.791 |

Cox proportional hazard regression was used to assess the association between an exacerbation in the year after each stepdown, compared with no stepdown, after adjusting for sex, age, BMI, smoking history, socioeconomic status, atopy, anxiety, depression, gastroesophageal reflux, and in the year prior to exposure: maximum ICS dose, ICS stability, ICS frequency, reliever frequency, LABA use, at least 2 add-on therapies, exacerbations, annual asthma review, inhaler technique check, and asthma management plan. The rate and number of exacerbations (per 10 person years, N) of patients that had an exacerbation by drug stepped down were none (1.23 per 10 person years, N = 9,984), ICS mean daily (0.84 per 10 person years, N = 2,163), ICS inhaler dose (1.13 per 10 person years, N = 720), LABA (1.18 per 10 person years, N = 458), other add-on (2.02 per 10 person years, N = 564). **Abbreviations:** BMI, body mass index; HR, hazard ratio; ICS, inhaled corticosteroid; LABA, long-acting beta-agonist.

Common ICS side effects, diabetes, and cataracts were not associated with initiating stepdown, but osteopenia/osteoporosis was significantly associated with not stepping down. Combination LABA–ICS use, an annual asthma review, past exacerbation, and being older were also significantly associated with not initiating ICS stepdown.

The factors significantly associated with stepping down add-on therapy included using ≥2 add-on therapies, LABA, past exacerbation, high-dose ICSs, older age, ICS stability, history of pneumonia, or arrhythmia (S6 Table).

## Cost savings

In the cohort of regular inhaler users, 31,379 patients were stepped down and had available drug costs in 2019. The mean annual saving was highest for LAMAs and LABAs and lowest for ICSs (ICS = £60.30 [95% CI 59.34–61.27], LABA = £126.92 [95% CI 123.61–130.22], LTRA = £108.86 [103.91–113.82], LAMA = £150.38 [139.65–161.12], theophylline = £114.98 [105.12–123.67]), S7 Table. Using the assumptions described in the methods, estimated UK savings from stepping down LABAs from half of the UK's stable adults treated with LABA–ICS would result in around 340,000 people having 7 fewer preventer prescriptions per year, saving approximately £17,000,000 annually. The equivalent calculations if stepping down ICSs would save around £8,600,000.

**Table 5. Risk of increase in reliever use by medication stepped down.**

| | Adjusted OR | 95% CI | p-Value |
|---|---|---|---|
| **Drug stepped down** | | | |
| None | *Reference* | | |
| ICS: inhaler dose | 0.99 | 0.98–1.00 | 0.594 |
| ICS: mean daily | 0.78 | 0.76–0.79 | <0.001 |
| LABA | 0.83 | 0.82–0.85 | <0.001 |
| Other add-on | 0.86 | 0.85–0.87 | <0.001 |

Logistic regression analysis was used to assess the association between an increase of one or more reliever inhaler prescription in the year after each stepdown, compared to no stepdown, after adjusting for sex, age, BMI, smoking history, socioeconomic status, atopy, anxiety, depression, gastroesophageal reflux, and in the year prior to exposure: maximum ICS dose, ICS stability, ICS frequency, reliever frequency, LABA use, at least 2 add-on therapies, exacerbations, annual asthma review, inhaler technique check, and asthma management plan. The number of patients that had an increase in reliever by drug stepped down was none (N = 19,027, 22.3%), ICS inhaler dose (N = 2,447, 39.6%), LABA (N = 1,269, 31.1%), other add-on (N = 832, 27.3%). **Abbreviations:** BMI, body mass index; ICS, inhaled corticosteroid; LABA, long-acting β2-agonist.

## Discussion

In a representative UK general asthma cohort, we have shown that between 2001 and 2017, prescription of higher-level preventer medications became increasingly common. There was often no clear clinical requirement for such high doses; one-third of patients had not received a reliever inhaler prior to their first asthma prescription or had an exacerbation. Nearly three-quarters of patients who were prescribed an ICS + 1 add-on as their first medication remained on the same medication for several years. One in 8 patients who were prescribed a medium/high-dose ICS as their first preventer already had known steroid-induced adverse effects. When medication stepdown was initiated, it was found to be safe, with no increase in exacerbations or reliever use, and economical.

In over 0.5 million asthma patients, the proportion prescribed higher-level medications (medium-dose ICSs, high-dose ICSs, or fixed-dose ICSs with add-on therapy) steadily increased over the past 2 decades. By 2017, only one-third of patients were managed on low-dose ICSs alone, in keeping with older UK studies also suggestive of inappropriate prescribing of higher doses of ICSs [28,29]. This pattern may have occurred because firstly, there was a relatively high number of patients, around one-third, who were prescribed their first preventer medication above the initial preventer step recommended by guidelines. Secondly, a high proportion of patients either remained on their incident prescription or had their medication escalated but never stepped down thereafter (including 77% of those prescribed ICSs + 1 add-on). Patients often have treatment stepped up in response to temporary worsening control or an acute exacerbation, but once control is achieved, their medication is not reviewed again [14]. Our study found most patients receiving a higher-level preventer medication as their first asthma medication had not received a prescription for a preventer inhaler in the previous year (guidelines pre-2016 advised reliever therapy alone as the first treatment step [30]) nor experienced an exacerbation. The prescribing of a higher level of medication without clear clinical indication suggests the prescribing physicians were not following guideline recommendations. This is perhaps not a surprising finding because adherence to guidelines in general practice has been shown to be low in many countries [28,29,31–34].

Physicians prescribing higher ICS doses were perhaps also not alert to potential adverse systemic effects because 13% of those prescribed medium- or high-dose ICSs as their first prescription already had known relevant comorbid conditions (i.e., diabetes, cataracts, glaucoma, osteopenia, or osteoporosis) compared with 11% of those prescribed low-dose ICSs. The majority of patients prescribed ICSs and an add-on were prescribed a fixed daily medium-dose or high-dose ICS–LABA combination inhaler. The latest recommendation in GINA for mild and moderate asthma, supported by multiple recent randomised control trials (RCTs) [35–38], is a low-dose ICS-containing inhaler. These latest guidelines should help encourage health professionals to prescribe the minimally effective treatment dose. Furthermore, for most patients, 80%–90% of the therapeutic benefit has been found to occur with low-dose ICSs [9,10].

The most common form of stepdown was halving ICS inhalations, a pragmatic approach that can be readily instigated by patients with or without direction from healthcare professionals. Stepdown did not significantly increase the risk of exacerbations, regardless of which medication was stepped down. Interestingly, there was a slight reduction in exacerbations when stepping down ICSs not seen when stepping down add-on therapies. This could be explained by confounding by indication from an unmeasured variable that led to patient selection only in patients with ICS overtreatment or could be due to increased adherence after medication change from increased medication awareness [39].

Findings remained robust even after all sensitivity analyses. As may be expected, the biggest positive confounding effect was a history of exacerbations in the year prior [15], but even when stratifying by this parameter, there was still no observed increase in exacerbations. This could be explained by patient selection towards the most stable patients, even those that had exacerbated. Another possibility is that this may reflect poor inhaler technique such that stepping down prescriptions did not significantly alter the absorbed dose. Considering stepping down is arguably still important in patients with low medication absorption (from either low adherence or poor technique) in terms of both reducing medication costs and alerting patients and physicians to these potential issues. Lastly, it is likely some patients were misdiagnosed [40], and routinely thinking about stepping down medication (as well as stepping up) as part of an annual asthma review could help identify such patients.

A Cochrane Review, 2017, looked at studies that stepped down ICSs in well-controlled asthma patients [19]. The review found 6 relevant studies; there was no difference in asthma exacerbations, asthma control, side effects, or quality of life. However, the quality of evidence was rated as low or very low. A Cochrane Review, 2015 [41], looked at studies that stopped LABA in well-controlled adults. Five studies were reviewed, with moderate quality of evidence. However, trials were short, with too few exacerbations to assess rates [41]. A small study published since found stopping LABAs was well tolerated [42].

There has only been one previous real-world study comparing stepped-down patients to controls. Using US administrative claims data, the authors identified a smaller cohort of 4,235 eligible patients with a shorter follow-up period and no exact timing of stepdown; this study found stepping down to be safe and highly cost-effective [43].

Around 1 in 6 controls had their treatment escalated in the year after the index date; this number was comparable to those that stepped down their ICS inhalations or their LABAs. A slightly higher number, around 1 in 5, escalated treatment if they had dropped LTRAs, LAMAs, or the ICS inhaler dose. Stepping down did not significantly increase the use of reliever medication, and most patients did not escalate treatment more than the controls; together, these findings suggest stepping down did not have a detrimental effect on patient's asthma control.

Although the reason for stepping down was not recorded, we were able to evaluate several potential prognostic factors. We found higher levels of medication and prescription stability were associated with instigating stepping down. Use of combination inhalers was associated with reduced odds of stepping down ICSs; however, this is likely to change with the latest GINA guidelines encouraging an as-needed approach. In contrast, common ICS adverse systemic effects, diabetes, cataracts, glaucoma, osteopenia, or osteoporosis were not associated with stepping down ICS. Older age was significantly associated with not having ICSs stepped down but was significantly associated with stepping down LABAs. Yet the older population are at the greatest risk of ICS side effects and are most likely to experience polypharmacy. In the UK, older people have a similar prevalence of asthma as younger adults but are the age group most likely to be prescribed ICSs and most likely to be prescribed medium- or high-dose ICSs instead [27]. It seems that age is a barrier to instigating stepping down, but our analysis found older age did not impact on asthma outcomes. Recently, there has been a drive in older people to rationalise medicines through deprescribing, and patients are keen to engage [44–46]. Notably, asthma-specific clinical practices, including annual reviews and management plans, did not appear to increase the chances of stepping down. A qualitative study of primary care health professionals suggested that key factors why stepping down does not occur are lack of confidence to step down, lack of time to discuss, and resistance from the patients [47].

Our cost analysis found considerable savings covering during just 1 year. The savings from dropping LABAs were almost 4 times that of ICSs and approximately double that of LTRAs. Scaling up the costs to estimate nationwide savings, assuming only half of all stable treated patients are stepped down, could potentially save around £17 million if LABAs were stepped down, equivalent to 2% of the UK's asthma budget, or £8.6 million if ICSs were stepped down. This large sum is because over 5.4 million people receive asthma treatment in the UK, with asthma medication contributing towards 13% of the total primary care prescribing costs [48] and ICSs being the second most prescribed medication [49]. Many patients prescribed higher doses have suboptimal inhaler technique, and hence, only a small proportion reaches the airways; the money saved on stepping down could instead be spent on improving and adapting education tools for patients and professionals, for example, to cover the continuously expanding inhaler market [50–52].

## Strengths and limitations

Major strengths were the use of a nationally representative study population, large sample size, and longevity of the data. Limitations of using routinely collected patient data include lack of information on daily medication use; therefore, to include patients who reduced the number of puffs, we had to calculate mean-daily ICS doses. This could have led to selection bias by excluding patients who decreased, then increased, their puffs again. However, our findings in this group were consistent with patients that had their ICS inhaler dose halved, suggesting this bias had a minimal influence on the effect estimate. It is possible there was selection bias from only including HES-linked patients in the stepping-down analysis because HES is only available for patients using English NHS hospitals. The nature of the data also did not allow us to know precisely why the decision to initiate stepdown was taken; hence, our analysis of potential prognostic factors attempted to investigate this. Although the algorithm used to identify asthma patients has 86% positive predictive value [17], a limitation of the data set is that it does not contain information on diagnostic tests such as lung function tests or airway hyperreactivity measures. Only around 4% of asthma patients are reviewed in secondary care, where such tests are performed in the UK [53]. Mild exacerbations treated only with an increase in ICSs could not be identified, but an annual increase in preventers and relievers was measured. Additionally, prescriptions in the exposed and unexposed may not have been dispensed or adhered to; however, our findings were comparable to the only other study using real-world data, which had access to dispensed data. Lastly, although guidelines suggest stepping down after 3 months of stability, 1 year of stability was used for the cost analysis; therefore, results may be more conservative than would truly occur.

## Conclusions

Over time, patients in the UK have been increasingly prescribed higher-level asthma medications with no clear clinical indication for needing higher doses. Worryingly, although stepping down of treatment is recommended by clinical guidelines, we found that it happened infrequently. Stepping down ICSs or add-on therapy did not appear to worsen health outcomes but did appear to result in significant cost savings. Medication burden seemed to be the main driver to stepping down; adverse medication effects were not. In summary, firstly, adverse treatment effects, medication burden, and cost to the health system should be considered when prescribing higher-level asthma medications for well-controlled asthma patients seen in primary care. Secondly, stepping down medication in stable asthma patients appears to be both safe and highly cost-effective.

## Supporting information

**S1 STROBE Checklist. STROBE checklist for observational studies for the study.**
(DOCX)

**S1 Table. Patient characteristics at their first asthma preventer prescription by level of medication prescribed, low-dose ICSs compared to higher-level treatment (medium/high-dose ICSs or ICSs + add-on therapy).** ICS, inhaled corticosteroid
(XLSX)

**S2 Table.** Sensitivity analysis, risk of exacerbation after stepdown (a) using a 6-month outcome window (instead of 12 months), (b) using a definition of stepping down starting 1 month or 3 months after the last prescription containing the stepped-down drug, (c) using an as-treated approach.
(XLSX)

**S3 Table. Effect of exacerbation history in year before stepdown.**
(XLSX)

**S4 Table. Time effect on first exacerbation of stepdown compared to no stepdown.**
(XLSX)

**S5 Table. Percentage of patients that escalated in the outcome year following stepdown.**
(XLSX)

**S6 Table. Factors found to be significantly associated with initiating stepdown, either ICSs or LABAs, ordered by strength of association.** Only factors significantly associated (p < 0.05) with initiating stepdown are shown (OR > 1); each variable is adjusted for all variables in the model. The effect estimate shown is the adjusted odds ratio as compared to their reference value. ICS, inhaled corticosteroid; LABA, long-acting β2-agonist.
(XLSX)

**S7 Table. Annual saving from stepping down each drug (including any escalation) in patients with drugs available in 2019, using 2019 NHS indicative prices.** NHS, National Health Service
(XLSX)

**S1 Fig. Incident asthma preventer prescriptions from across the UK, 2001–2017.** Add-on therapy refers to LABA, LTRA, theophylline, or LAMA. ICS, inhaled corticosteroid; LABA, long-acting β2-agonist; LAMA, long-acting muscarinic antagonist; LTRA, leukotriene receptor antagonist
(TIF)

**S1 Study Protocol. Study protocol for the study.**
(DOC)

**S1 CPRD medcodes. CPRD codes for the variables included in the study.** CPRD, Clinical Practice Research Datalink
(PDF)

## Author Contributions

**Conceptualization:** Chloe I. Bloom.

**Data curation:** Chloe I. Bloom.

**Formal analysis:** Chloe I. Bloom.

**Methodology:** Chloe I. Bloom, Laure de Preux.

**Supervision:** Aziz Sheikh, Jennifer K. Quint.

**Writing – original draft:** Chloe I. Bloom.

**Writing – review & editing:** Laure de Preux, Aziz Sheikh, Jennifer K. Quint.

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
