## [Editor Report · Decision Letter 0]

10 Jan 2020

Dear Dr Bloom, 

Thank you for submitting your manuscript entitled "Health and cost impact of stepping-down asthma medication: a UK population-based observational study" for consideration by PLOS Medicine.

Your manuscript has now been evaluated by the PLOS Medicine editorial staff [as well as by an academic editor with relevant expertise] and I am writing to let you know that we would like to send your submission out for external peer review.

**Please be aware that, due to the voluntary nature of our reviewers and academic editors, manuscript assessment may be subject to delays during the holiday season. Thank you for your patience.**

Kind regards,

Caitlin Moyer, Ph.D.

Associate Editor

PLOS Medicine

---

## [Decision Letter · Decision Letter 1]

16 Mar 2020

Dear Dr. Bloom,

Thank you very much for submitting your manuscript "Health and cost impact of stepping-down asthma medication: a UK population-based observational study" (PMEDICINE-D-20-00031R1) for consideration at PLOS Medicine. 

Your paper was evaluated by an academic editor with relevant expertise and sent to independent reviewers, including a statistical reviewer. The reviews are appended at the bottom of this email and any accompanying reviewer attachments can be seen via the link below:

[LINK]

In light of these reviews, we will not be able to accept the manuscript for publication in the journal in its current form, but we would like to invite you to submit a revised version that fully addresses the reviewers' and editors' comments. You will appreciate that we cannot make a decision about publication until we have seen the revised manuscript and your response, and we expect to seek re-review by one or more of the reviewers. 

We hope to receive your revised manuscript by Apr 06 2020 11:59PM. Please email us (plosmedicine@plos.org) if you have any questions or concerns.

Please let me know if you have any questions. Otherwise, we look forward to receiving your revised manuscript in due course. 

Sincerely,

Richard Turner PhD, for Caitlin Moyer, Ph.D.

Associate Editor, PLOS Medicine

rturner@plos.org

Please quote the study dates in the title.

In your abstract and throughout the paper, please add p values alongside 95% CI where available. 

We ask you to quote aggregate demographic details for study participants in the abstract. 

Please add a new final sentence to the "Methods and findings" subsection of your abstract, quoting 2-3 of the study's main limitations. 

At line 54, please begin the sentence with "In this study, we observed that ..." or similar. 

After your abstract, please add a new and accessible "author summary" section in non-identical prose. You may find it helpful to consult one or two recent research papers published in PLOS Medicine to get a sense of the preferred style. 

Early in the methods section of your main text, please state whether your study had a protocol or prespecified analysis plan (we note that you mention a protocol at line 204) and, if so, attach the document(s) as a supplementary file, referred to in the text. Please highlight analyses that were not prespecified. 

Please restructure the early part of your discussion section so that the first paragraph provides a summary of the study's findings. 

Please make that "include" at line 370. 

Throughout the paper, please quote exact p values or "p<0.001".

Throughout the text, please format reference call-outs as follows: "... people with asthma [1,2].".

In your reference list, please remove the academic editor's name from reference 4. 

Please attach a completed checklist for the most appropriate reporting guideline, which may be STROBE, as a supplementary document (referred to in your methods section). In the checklist, please refer to individual items by section (e.g., "Methods") and paragraph number rather than by line or page numbers, as the latter general change in the event of publication. 

Comments from the reviewers:

*** Reviewer #1: 

"Health and cost impact of stepping-down asthma medication: a UK population-based observational study" finds that asthma patients were bring prescribed higher levels of treatment oft without clear clinical indications, from nationwide UK primary care medical records from 2001 to 2017. Further analysis concluded that stepping-down medication to the minimum effective dose did not adversely affect outcomes, and moreover saved on costs. Indeed, stepping-down of ICS in particular consistently reduced hazard ratios (Tables S2/S3).

A major strength of this study would lie in the large number of patients involved (over 100,000 patients for each of the incident prescription/de-escalation analyses), from a cohort of over 500,000 patients from the UK primary care system over 17 years. Findings on the efficacy of stepping-down remained largely unchanged under relevant sensitivity analyses. To the best of our knowledge, there exists no prior study of comparable scale on the impacts of stepping-down asthma medication. Therefore, this study may have significant potential to affect health policy on asthma management, especially with an observed increase in the prescription of higher level medication, and purported insufficient evidence about stepping-down ICS medication as a factor in increased risk of future asthma attacks (e.g. from page 33 of the BTS/SIGN 2019 guidelines).

There however remain some points that might be clarified:

1. In lines 62-64, The appropriate citations might be provided for the various asthma guidelines (GINA, the Expert Panel Report, BTS/SIGN)

2. From line 126, how was the criteria of >= 3 ICS prescriptions (parallel to infrequent ICS defined as <= 2 prescriptions per year in line 166) decided upon, given the mean of 7 prescriptions per year (line 133)? Data on the distribution of prescriptions/year might be illuminating.

3. On the definition of "stepping-down" (Line 138 onwards) - some possible configurations of medication step-down appear not to be explicitly covered by the chosen definition. For example, in prior work ("Stepping Down Asthma Treatment: How and When", Rogers & Reibman, Curr Opin Pulm Med. 2012 Jan; 18(1): 70-75.), it is claimed that "certain step-down practices such as the complete cessation of ICS to LABA alone was associated with a significant loss of asthma control". How might dropping ICS to an add-on therapy be categorized in the analysis (e.g. in Table 4 & 5)?

4. More generally, how were the stepping-down definitions decided? Is halving dosage, for example, universally accepted practice as opposed to say a reduction of dosage by 25% (it is stated in line 309 that "The most common form of step down was halving ICS inhalations", but this does not seem to preclude other practices)? 

5. The incidence rate (N) for the various drug-stepped-down categories might be provided in Tables 4 & 5, for convenience.

6. The definition of reliever prescriptions as a secondary outcome (e.g. line 154) might not be entirely obvious. The authors might consider adding a short description of preventer/reliever usage practices in the Introduction. Further, what is the tradeoff between reduced preventer and increased reliever usage?

7. On the incident analysis, it was stated that (line 229) "Of those prescribed higher level medication as their incident asthma prescription... 48.4% had not excerbation or reliever prescriptions in the previous year (Table 2)"; however, the 48.4% figure is not immediately evident from the presentation of Table 2.

8. Moreover, the authors might consider standardizing the formatting of Tables 2/S1/S4 (i.e. consistently place "N" and "%" as separate columns), and include p-values where appropriate.

9. The characteristics examined for incident vs. step-down analysis (i.e. Table 2 vs. Table 3) are fairly different; for instance, age and socioeconomic status are not considered in the incident analysis, although it is not obvious that they do not affect/correlate with incident prescriptions. The authors might consider briefly discussing the selection of characteristics for each analysis.

10. In line 236, it is stated that "(of) 125,341 patients, 39,897 were stepped-down, and 85,460 were not stepped down". However, the sum of stepped-down and non-stepped-down patients appears to be 125,357 patients.

11. Also, from the definition of stepping-down and the chronological length of the study (17 years), it seems possible that an individual (exposed) patient might step-down (and escalate) medication multiple times, over the full time period. How are such cases, if any, accounted for?

12. From line 292, "Our study found most patients receiving a higher level preventer medication as their first asthma medication had not received a prescription for a preventer inhaler in the previous year (guidelines pre-2016 advised preventer therapy alone as the first treatment step(23))", is "preventer medication" not considered "preventer therapy"?

13. In line 310, it is mentioned that step-down can be instigated by patients, possibly without direction from healthcare professionals. How does this happen in practice? Moreover, for the prescription data used in the analyses, were they taken as prescribed by the primary care health professionals (i.e. possibly not followed up on), or as filled by pharmacists?

14. In line 355, which two studies do "both studies" refer to?

15. The discussion on prior investigations on medication step-down for asthma may be somewhat perfunctory. For example, it is claimed that (line 328) "Several randomised controlled trials of medication step-down have shown no difference in asthma exacerbations, asthma control, side-effects or quality of life"; however, citation (36) alone mentions that "Of 14 randomized controlled studies on step down of asthma medications (follow-up period range, 12-168 weeks), only 4 of them found no significant differences in asthma outcomes compared with patients maintaining their asthma medications". 

Therefore, while there appears to be a general consensus from the latest asthma management guidelines on stepping-down controlled asthma cases to a minimal level, the evidence from previous randomized controlled trials appears somewhat mixed, even considering recognized methodological limitations. The authors might consider commenting in greater depth on these previous studies.

16. Finally, there may be some minor grammatical issues, e.g.

Line 39: "A cohort of patients were drawn..." -> "was drawn"

Line 47: "...and no increased in reliever" -> "no increase"

Line 297: "Perhaps not a surprising finding..." -> "This is perhaps not a surprising finding..."

Line 337: "this number was comparative..." -> "was comparable"?

*** Reviewer #2: 

This is an excellent study showing that asthma patients in the UK (1) are often initiated on overly intensive therapy, (2) are kept on intensive therapy without attempts to step down (contra guidelines), and (3) have no better outcomes than those whose therapy is stepped down.

Major

1. My biggest concern is with the third finding—that stepping down therapy was not associated with more exacerbations or SABA use than continuation of therapy. As the authors recognize, there is a strong possibility for selection bias. They perform a stratified analysis based on exacerbations in the prior year. They also report in Tables 4 and 5 regression analyses that control for a number of potential confounders. These steps all help address the worry of unmeasured confounding. However, they did not perform the analysis with propensity score matching (or high-dimensional propensity score matching), which may help further allay concerns. In general, I worry that the group whose therapy was stepped down is different in important ways than the group whose therapy was continued.

2. The authors did not report on any difference in the side-effects (diabetes, cataracts, glaucoma, osteopenia, or osteoporosis) in the group of patients who were stepped down compared to those who were not. Was there any improvement in these variables among those who were stepped down (whether fewer new diagnoses or improvement in symptoms [e.g. HgbA1c])?

3. This may be beyond the scope of data, but I wonder what proportion of patients had true asthma. As the authors allude to in the discussion (citing the paper by Aaron SD et al., JAMA 2017), many patients who carry a diagnosis of asthma and are started on inhalers never truly have asthma. What proportion of the patients in the cohort had PFTs or a methacholine challenge confirming a diagnosis of asthma? It is possible, as the authors acknowledge, that many patients did not have asthma—in which case, we would expect that stepping down medications would have no effect. Conversely, perhaps those with true asthma may have had more exacerbations and SABA use after being stepped down. 

4. The authors note that, in the 107,908 patients in incident cohort, one-third were on an ICS and add-on, of which 94% were on a LABA. What proportion were on combination ICS-LABA versus two separate inhalers (ICS and LABA)? It would be interesting to examine whether being on a combination product affects the likelihood of being stepped down.

5. Of the 107,908 patients who were on an ICS-LABA combination, how many were on ICS-formoterol? The approach of using Symbicort as needed for mild asthma is new in the guidelines. But I wonder whether some clinicians in the UK were already doing this, in which case some of those on an ICS-LABA may not be getting stepped down because they are already on a low intensity therapy (i.e. ICS-formoterol PRN).

Minor

Line 77: This should be "is obtained with a low-dose" rather than "is obtained with low-dose."

Line 110: This should be: "Prescription data were…" rather than "prescription data was..."

Line 121: Does excluding non-HES linked patients affect the generalizability? Is there any reason to think that non-HES linked patients are different from HES-linked patients? This may be worth commenting on in the limitations.

Line 286: This should be "…a higher number of patients, around one-third, who were prescribed…" rather than "a higher number of patients, around one-third, that were prescribed…"

Line 297: This should be "Perhaps this is not a surprising finding…" rather than "Perhaps not a surprising finding…"

Line 373: This should be: "…excluding patients who decreased…" rather than "…excluding patients that decreased…"

*** Reviewer #3: 

This is an analysis of the Clinical Practice Research Datalink (UK). The authors used observational data to describe asthma medication patterns with a focus on health outcomes and costs of stepping down asthma medications. The primary conclusion is that higher doses and numbers of asthma medications were prescribed in the UK from 2001-2017, and that most patients can safely reduce them, at a high cost savings.

Major comments:

1-One of the most interesting questions related to step down that was not considered in this analysis is listed as an asterisk in Table 1 (patients that stop all preventer meds were not included). Why not? I could not find in the methods section where this choice is explained. Stopping all controller meds (especially ICS) is probably the highest risk step down (see, for example, recent GINA guideline summarizing these risks). Also, it is potentially very relevant to new recommendations which suggest all asthma patients should use some level of ICS (intermittent or continuous). Without information on this step down risk, it is difficult to broadly state that step down moves do not adverse affect outcomes. I think the authors either need to analyze this group or be more precise in describing which step down types are likely low risk, and which were not analyzed.

2-I think the following variables should be defined more clearly, as in how they were identified within the dataset and what they mean: "asthma management plan, annual asthma review, and inhaler technique check." I also wonder if having seen a specialist is another variable that should be included. 

3-The final concluding statement (lines 391-394) which follows "In summary" focuses on not stepping up (which was a descriptive trend finding) and I think most of the analysis is related to outcomes of stepping down. Therefore, the authors may consider a summary statement that includes stepping down.

4-How to accomplish stepping down (i.e. how to solve this problem) is not clear from their data. Having annual visits, checking inhaler technique, and an asthma management plan do not seem to be related to step down. Are there any patterns from their data that suggest a solution? Are there signs of stepping down to ICS PRN and that this could be a successful strategy?

Other comments:

1-Introduction section is concise and clear.

2-Sensitvity analyses are appropriate

3-Line 154-155--is there validation or a reference to justify 1 or more relievers as cut off?

4-Supplementary Figure 2 is very helpful. I suggest the authors consider moving this to the main document.

***

[LINK]

---

## [Decision Letter · Decision Letter 2]

1 May 2020

Dear Dr. Bloom,

Thank you very much for re-submitting your manuscript "Health and cost impact of stepping-down asthma medication: a UK population-based observational study, 2001 - 2017" (PMEDICINE-D-20-00031R2) for consideration at PLOS Medicine.

I have discussed the paper with editorial colleagues and our academic editor, and it was also seen again by three reviewers. I am pleased to tell you that, provided the remaining editorial and production issues are fully dealt with, we expect to be able to accept the paper for publication in the journal.

[LINK]

Please let me know if you have any questions. Otherwise, we look forward to receiving the revised manuscript shortly. 

Kind regards,

Richard Turner, PhD

rturner@plos.org

Requests from Editors:

Please note in your competing interest statement that AS is a member of PLOS Medicine's Editorial Board. Also, please add a few words to explain whether CIB has received grant funding, for example, from AstraZeneca etc.

Your data statement mentions that "code lists will be provided on request", which I'm afraid is not consistent with PLOS Medicine's data policy. Please provide these in a supplementary document to be published with the manuscript. An alternative would be to deposit these in a public repository.

We ask you to adapt the title slightly to: "Health and cost impact of stepping-down asthma medication for UK patients, 2001 - 2017: a population-based observational study".

Around line 45, please quote the number or proportion of patients stepped-down.

At line 52 in your abstract and all other instances, please quote p<0.001 or exact values. 

At line 59, please make that "Study limitations included the possibility that prescribed medication ..." or similar.

At lines 62 and 472, please adapt the text to "patients in the UK", or similar to indicate where the study was done. 

At line 91, please remove "this one of the largest studies".

At lines 134-135, please add call-outs to the attached protocol and STROBE documents (e.g., "See S1_STROBE_Checklist").

Please make that "data are" at line 137.

Please remove the copyright information at line 143 (the sentence can be truncated at "... within the CPRD data.".

Please substitute "sex" for "gender" where appropriate (e.g., in the legend for table 4). 

Please add full access details to reference 26, and any other references that lack it. 

Please remove the header "Page number" from the rightmost column of your attached STROBE checklist, and the numbers below if these are page numbers (which generally change in the event of publication). Please restructure this column to provide section details (e.g., "Methods") and paragraph numbers.

Comments from Reviewers:

*** Reviewer #1: 

The authors have adequately responded to all points previously raised, in their response. However, it seems that some line numbers referred to in the author response may have shifted (perhaps after further editing as additional points were addressed?); in particular, the response for Point 4 on "most randomised controlled trials used 50% step-down and a larger step-down would be more likely to identify an effect if there as one" was not readily encountered in the text. Other than these, there are no further concerns on our part.

*** Reviewer #3: 

The authors are responsive to reviewer feedback. Overall, the methods, interpretation, and communication of the findings are excellent. 

A few additional points to consider:

1. The propensity score analysis strengthens the conclusion. I was able to find a description of the methods for the PS, but I did not find the PS in Table S5 as suggested in the authors' responses. I could not find this info listed in a table. I apologize if I missed this. Can the authors please double check to confirm they included this information? 

2. I disagree with the authors' decision not to consider the outcomes when patients completely stop ICS, an event that is more likely to result in increased exacerbation than a decrease in controller meds (high level evidence from meta-analysis). Seeing this group and showing a negative outcome would increase the validity of using the dataset for outcome assessment. The argument that these patients are judged not to have asthma is an assumption that may hold for a fraction of patients who completely stop controller medications. However, many others likely stop on their own (without provider direction) or revert back to SABA prn approaches, an accepted GL management option for intermittent asthma prior to GINA 2019 GL release. 

3. I agree with the choice not to analyze step down to LABA monotherapy, as this is not an accepted or GL recommended asthma therapy.

***

[LINK]

---

## [Editor Report · Decision Letter 3]

9 Jun 2020

Dear Dr Bloom, 

On behalf of my colleagues and the academic editor, Dr. Aaron Kesselheim, I am delighted to inform you that your manuscript entitled "Health and cost impact of stepping-down asthma medication for UK patients, 2001-2017: population-based observational study" (PMEDICINE-D-20-00031R3) has been accepted for publication in PLOS Medicine. 

PRODUCTION PROCESS

PRESS

PROFILE INFORMATION

Thank you again for submitting the manuscript to PLOS Medicine. We look forward to publishing it. 

Best wishes, 

Richard Turner, PhD

Senior Editor 

PLOS Medicine

plosmedicine.org